# Identification of Histopathological Criteria for the Diagnosis of Canine Cutaneous Progressive Angiomatosis

**DOI:** 10.3390/vetsci9070340

**Published:** 2022-07-05

**Authors:** Francesca Abramo, Marta Vascellari, Giada Colzi, Luca Pazzini, Francesco Albanese, Lara Olivieri, Claudia Zanardello, Claudia Salvadori, Giancarlo Avallone, Paola Roccabianca

**Affiliations:** 1Department of Veterinary Sciences, University of Pisa, 56124 Pisa, PI, Italy; giada.colzi@unipi.it (G.C.); claudia.salvadori1819@gmail.com (C.S.); 2Histopathology Department, Istituto Zooprofilattico Sperimentale delle Venezie, Viale dell’Università 10, 35020 Legnaro, PD, Italy; mvascellari@izsvenezie.it (M.V.); czanardello@izsvenezie.it (C.Z.); 3MYLAV Veterinary Laboratory La Vallonea, 20017 Passirana di Rho, MI, Italy; lucapazzini@laboratoriolavallonea.net (L.P.); vetfra1@yahoo.it (F.A.); 4Ospedale Veterinario I Portoni Rossi-Anicura, 40069 Zola Predosa, BO, Italy; lara.olivieri@anicura.it; 5Department of Veterinary Medical Sciences, Università di Bologna, Via Tolara di Sopra 50, 40064 Ozzano dell’Emilia, BO, Italy; giancarlo.avallone@unibo.it; 6Department of Veterinary Medicine and Animal Sciences, Università degli Studi di Milano, Via Celoria 10, 20133 Milano, MI, Italy; paola.roccabianca@unimi.it

**Keywords:** progressive angiomatosis, vessels, endothelium, dog, skin

## Abstract

**Simple Summary:**

In animals, cutaneous progressive angiomatosis (CPA) is a disorder with variable prognosis related to the extension and depth of infiltration of the surrounding tissues by vessels. CPA may share some microscopical features with other vascular proliferations such as low-grade well-differentiated capillaritic hemangiosarcoma (HS), making the diagnosis not always straightforward. The aim of this study is to assess the most common diagnostic microscopical features of CPA in dogs. Eleven histopathological criteria were analyzed on 31 CPA and 11 primary cutaneous HS in dogs. Features significantly associated with CPA included: lobular growth, interposition of connective tissue and adnexa between the vascular proliferation, presence of nerve fibers, and a mixed vascular proliferative component. Absence of plump/prominent endothelial cells, lack of atypia, and lack of mitoses were also significant factors for differentiating CPA from HS. In conclusion, the presence and/or absence of multiple microscopical clues allowed for the distinction of CPA from HS.

**Abstract:**

The term angiomatosis is used to denote a group of well-known to poorly characterized proliferative vascular entities. In animals, cutaneous progressive angiomatosis (CPA) is a disorder with variable prognosis related to the extension and depth of infiltration of the surrounding tissues by vessels. CPA may share some microscopical features with other vascular proliferations such as low-grade well-differentiated capillaritic hemangiosarcoma (HS), making the diagnosis not always straightforward, especially in small biopsies. The aim of this study is to retrospectively assess the most common diagnostic microscopical features of CPA in dogs. In this work, 11 histopathological criteria were analyzed on 31 CPA and 11 primary cutaneous HS in dogs. Features significantly associated with CPA included: lobular growth, interposition of connective tissue and adnexa between the vascular proliferation, presence of nerve fibers, and a mixed vascular proliferative component. Absence of plump/prominent endothelial cells, lack of atypia, and lack of mitoses were also significant factors differentiating CPA from HS. Additional distinctive findings in CPA, although with no statistical association to CPA diagnosis, were vascular shunting, absence of necrosis, and endothelial cell piling up. In conclusion, the combined use of different microscopical clues allowed for the distinction of CPA from HS and was considered useful for the diagnosis of CPA.

## 1. Introduction

Angiomatosis encompasses a miscellaneous group of poorly characterized vascular entities comprising complex benign (possibly malformative) to aggressive (progressive forms) vascular proliferations. Canine angiomatosis was first described by Peavy et al. [1] and was more extensively characterized by Gross et al. [2] and Roccabianca et al. [3]. However, comprehensive microscopical descriptions and complete case reports are scarce [4,5,6,7,8,9,10]. Gross et al. [2] separate cutaneous progressive angiomatosis (CPA) from vascular hamartomas and angiomatosis secondary to lymphedema. On the contrary, the classification of angiomatosis in human medicine encompasses vascular malformations and vascular reactive lesions, with the latter potentially regressing spontaneously, while malformative lesions require surgery [11]. In small animals, CPA is defined as an infiltrative, but non-neoplastic, vascular proliferation [2,10] thus, differing from the human counterpart, by a progressive behavior with prognosis depending on the extension, depth and localization of the infiltration of tissue invasion. In the recently revised international classification of Tumors of Soft Tissue [3], angiomatosis in animals has been further subdivided according with morphology into cavernous, capillary, and arteriovenous types, and some entities, such as progressive angiomatosis in small animals, have been better delineated. [3]. Histologically, CPA is described as an infiltrative lesion composed of variable combinations of vessels of different sizes and types, comprising variable numbers of arteries, veins, and lymphatics, but with a consistent representation of capillaries that may predominate or not. Additional features variably present in CPA include the presence of clusters of vessels around a central (feeding) malformed central vessel, anastomosing, sinusoidal spaces, endoluminal papillary projections, and phalangeal bone infiltration. Arteriovenous shunting may be present and is deemed as one of the possible causes of disease progression. Endothelial cells of CPA are usually flat, in single layer, and atypia and mitoses are generally missing. [2] Due to the histologically bland appearance but the infiltrative growth, ruling out vascular malformations or a low-grade capillary angiosarcoma (HS) may be challenging, especially in small punch biopsies.

The aim of this retrospective study is to assess microscopically and apply the most recent classification [3] to cases of putative CPA and to attest the microscopical features most useful for CPA diagnosis. Furthermore, nerve fibers were assessed within the lesions, as this is considered a diagnostic clue in humans [12] but seems to have never been mentioned nor investigated for small animal CPA.

## 2. Materials and Methods

### Sample Collection, Histopathology and Statistical Analysis

Formalin fixed, paraffin embedded skin biopsies from dogs with a diagnosis of CPA were retrospectively retrieved from the histopathology archives of four diagnostic services. Cases of scrotal-type hamartomas were excluded and cases of angiomatosis secondary to lymphedema were never retrieved. For each case, four-micrometer-thick tissue sections were cut, routinely stained with H&E, and reviewed by a pathologist with dermatopathology expertise (FA, PR) and a neuropathologist (CS). The following histopathological criteria were evaluated: (1) site of lesions, superficial versus deep tissue involvement (dermis only, dermis and subcutis or subcutis only); (2) expansive vs. infiltrating growth; (3) growth pattern (papillary endovascular, lobular, solid); (4) evidence of normal tissue components interposed among CPA components; (5) type of vessels represented and predominance of vessel type, namely arteries, veins, capillaries, lymphatics; (6) cytological features (endotheliocyte shape, piling up, mitotic count assessed in a 2.37 mm^2^ area, atypia); (7) presence/absence of vascular shunting; (8) presence/absence of malformed/misshapen vessels; (9) presence /absence of possible secondary lesions: thrombosis and necrosis; (10) microscopical evidence of bone invasion (indicative of aggressive lesions); (11) presence of nerve fibers. Eleven cases with a diagnosis of cutaneous HS were included for comparison. The number of masses on each dog was recorded at time of diagnosis. Data on signalment, presentation, and follow up were recorded when possible (Table 1). Follow-up information through phone interviews with referring veterinarians was collected when possible.

Association of diagnosis (CPA vs. HS) with the histological variables was statistically assessed by Chi-square test.

## 3. Results

### 3.1. Clinical Presentation

Thirty-one cases with a CPA diagnosis were initially collected; of these, three were excluded because histological features were consistent with a spindle cell hemangioma, an infiltrative hemangioma, and a hemangiosarcoma. The remaining 28 cases were consistent with progressive angiomatosis and were included in the study. Signalment and presentation are listed in Table 1. Age was available in 25 cases, with a range of 2 months–13 years and a mean age of 6 years. There were 7 females and 19 males; in 2 cases the sex was unknown. There were 13 mixed breed dogs while pure breeds included Dachshund (2), Dogue de Bordeaux (2) and one dog for each of the following: Australian sheepdog, Boxer, Bull mastiff, Cocker spaniel, German shepherd, Great Dane, Husky, Pincher, Shorthaired pointer, Springer spaniel and Yorkshire terrier. Out of 28 cases, 12 lesions were in the limbs, 7 were in the head, and 6 were in the trunk, while in three cases, the anatomical site was not recorded. At initial diagnosis, five dogs had serpiginous or multiple grouped lesions, nine lesions were erythematous to purple and five were variably ulcerated (Figure 1a–d). Duration of lesions at time of diagnosis was recorded in 13 cases: in 2 dogs the lesion was observed early at the onset; in 6 cases time of duration ranged from 1 to 4 months prior to biopsy; for the other 3 cases, lesions were present for a long but unspecified time. Due to the retrospective nature of this report, follow-up information was retrieved from only five dogs. Of these, four underwent excisional surgery and one was treated with photo laser; only one case recurred and amputation was performed two times, as a consequence of bone invasion, but the dog was still alive at the end of the study; at the end of the study, three dogs had deceased for causes unrelated to the vascular lesion.

### 3.2. Histopathology

Histopathological criteria assessed for CPA and HS are listed in Appendix A respectively.

•Criterion 1. In most CPA cases, in the deep dermis and subcutis, increased vessel density was observed (17/28, 61%); 4/28 (14%) cases had only dermal lesions and 7/28 (25%) had lesion only in the subcutis. HS developed in the dermis in 5/11 (45%), in the subcutis in 4/11 (36%) and at both sites in 2/11 cases (18%).•Criterion 2. All cases of CPA (100%) had infiltrative growth. Most HS (8/11, 73%) had infiltrative growth as well.•Criterion 3 (Figure 2a–c). Eight cases of CPA (29%) had multiple large vessels with endoluminal papillary growth; in three of these, concurrent lobular growth was observed; in 20 cases (71%) lobular vascular growth pattern was seen, while a solid pattern of proliferation was never detected. All HS were characterized by irregular blood-filled channels and lacunae, in 2 (18%) a concurrent lobular growth was observed, solid/epithelioid areas within the tumors were seen in 6/11 cases (55%).•Criterion 4 (Figure 2d–f). In all cases of CPA (100%), normal tissue was interposed between vascular lesions. Normal tissue components varied in type and amount according with the lesion site and consisted of adnexal appendages (follicles, sebaceous and sweat glands, *pili erector* muscle) and structured connective tissue, fat and/or skeletal muscles. In HS, interposed normal tissue was seldom observed in the core of lesions (3/11, 30%).•Criterion 5 (Figure 3a–c). In CPA, variable types and variable combination of vessel types were observed and included capillaries, arteries, veins and lymphatics. Presence of all vessel types was seen in 8/28 (29%) cases; all hematic but not lymphatic vessels was detected in 16/28 (57%) cases; in four cases, arteries or veins and capillaries were seen. According to prevalence of the vessel type, cases of CPA were classified as arteriovenous (12/28, 43%), capillary (9/28, 32%), mixed cavernous with either a capillary or an arteriovenous component (5/28, 18%), or mostly cavernous (2/28, 7%). For HS, all channels and lacunae within the neoplasia were interpreted as newly formed abnormal vascular structures composed of neoplastic endothelial cells, with occasional vasoformation consisting of dilated capillaries and veins (100%).•Criterion 6 (Figure 3d–f). The endothelial lining of vessels in CPA was flat for all cases (100%); in 4/27 (15%) cases nuclei were focally plump and prominent but without atypia; piling up as well as mitoses were never detected. In all HS, the endothelial lining was prominent and characterized by cytological atypia (100%), in 2/11 (18%) cases, piling up was detected; median mitotic count was 11 in a 2.37 mm^2^ area (range 4–58).•Criteria 7 & 8 (Figure 4a–c). In CPA, arteriovenous shunts were seen in 4/27 cases (15%) and malformed vessels were present in 14/28 (48%) cases; none of the aforementioned alterations were observed in HS.•Criterion 9. In CPA, thromboses were occasionally detected (7/28, 25%) in medium to large sized vessels, while necrosis was never observed; in HS, thromboses and necrosis were seen in three and two cases, respectively (27%, 18%).•Criterion 10 (Figure 4d). Bone invasion was observed in 1/28 cases (4%), with digit involvement; bone invasion was never recorded in HS cases.•Criterion 11 (Figure 4e,f). Nerve bundles were often seen at the vessel periphery in a large proportion of CPA cases (20/28, 71%). Intralesional nerve bundles were detected in CPA as small central axons covered by myelin sheaths and ensheathed by endonevrium. Nerve bundles were never detected within HS.

**Figure 2 vetsci-09-00340-f002:**
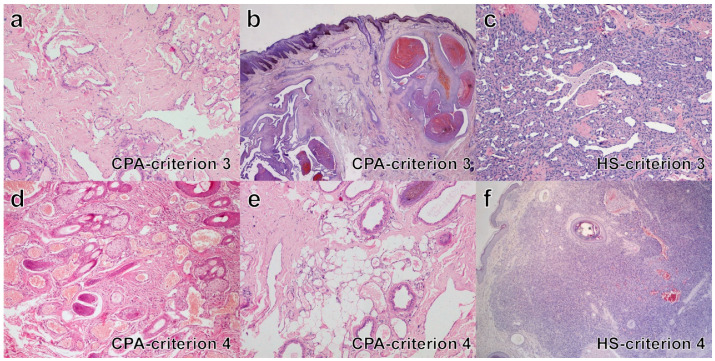
Histopathology in cases of CPA and HS (H&E stain): criteria 3 and 4. (**a**) CRITERION 3, CPA case n. 11: multifocal, lobular vascular growth pattern; (**b**) CRITERION 3, CPA case n. 16: multiple large vessels with endoluminal papillary growth; (**c**) CRITERION 3, HS case n. 10: irregular blood-filled channels and lacunae; (**d**) CRITERION 4, CPA case n. 1: interposed adnexal appendages (follicles, sebaceous glands) between vessels; (**e**) CRITERION 4, CPA, case n. 11: interposed fat and apocrine glands between vessels; (**f**) CRITERION 4, HS, case n. 2: only one follicle within the core of the lesion.

**Figure 3 vetsci-09-00340-f003:**
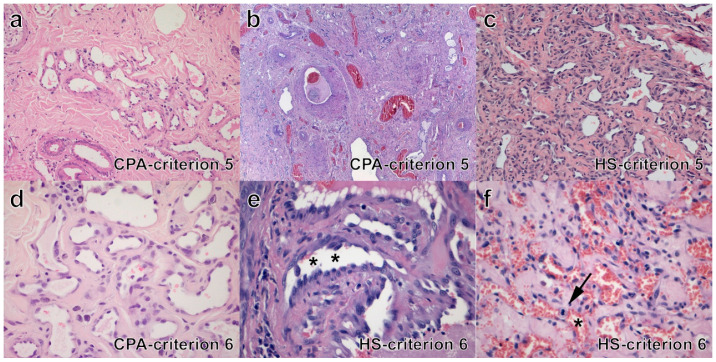
Histopathology in cases of CPA and HS (H&E stain): criteria 5 and 6. (**a**) CRITERION 5, CPA, case n. 11: combinations of small veins, arteries and lymphatics; (**b**) CRITERION 5, CPA, case n. 27: combination of different type and sized vessels; (**c**) CRITERION 5, HS, case n. 10: channels of newly formed abnormal vascular structures; (**d**) CRITERION 6, CPA, case n. 3: flat and only focally plump endothelial lining; (**e**) CRITERION 6, HS, case n. 11: prominent endothelial lining with occasional piling up (asterisks); (**f**) CRITERION 6, HS, case n. 3: cytological atypia of the endothelial lining and one mitotic figure (asterisk).

**Figure 4 vetsci-09-00340-f004:**
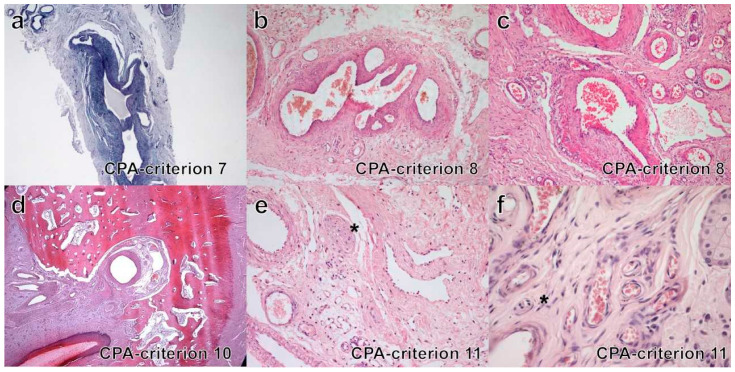
Histopathology in cases of CPA and HS (H&E stain): criteria 7, 8, 10 and 11. (**a**) CRITERION 7, CPA, case n. 6: arteriovenous shunt; (**b**) CRITERION 8, CPA, case n. 5: malformed vessel; (**c**) CRITERION 8, CPA, case n. 23: malformed vessel; (**d**) CRITERION 10, CPA, case n. 28: bone invasion; (**e**) CRITERION 11, CPA, case n. 14: intralesional nerve bundle (asterisk); (**f**) CRITERION 11, CPA, case n. 15: intralesional nerve bundle (asterisk).

### 3.3. Statistic Analysis

Histological features significantly associated with the diagnosis of CPA were lobular growth (*p* < 0.001), presence of mixed vessel types (*p* < 0.001), presence of arteries (*p* < 0.001), veins (*p* < 0.001), and lymphatics (*p* = 0.04), evidence of interposed connective tissue (*p* < 0.001) and adnexa (*p* = 0.004) and presence of nerve bundles at the periphery of the lesion (*p* < 0.001). Histological features significantly associated with the diagnosis of HS were presence of channels/lacunae (*p* < 0.001) and solid growth (*p* < 0.001), prominent endothelial lining (*p* < 0.001), endothelial cell atypia (*p* < 0.001) and presence of mitoses (*p* < 0.001). 

The other variables showed no statistically significant association with the diagnosis.

## 4. Discussion

In this study, morphological criteria specific for CPA diagnosis were assessed and some criteria were found to be significantly correlated with a diagnosis of CPA or HS. Morphological features significantly associated with CPA consisted in the finding of lobular growth, mixed vascular composition with or without misshapen vessels, and the presence of normal interposed tissue components and nerve fibers between vascular proliferations. Noteworthy, atypia, mitotic activity, and prominence of endothelial cells were absent in CPA, but were significantly associated to the diagnosis of HS. Additional distinctive findings, although not statistically associated with CPA diagnosis included vascular shunting, lack of necrosis, and lack of endothelial cell piling up. Thus, the combination of different morphological features results were useful and aided in the diagnostic for CPA against cutaneous HS.

Additional features observed in canine CPA, as well as in HS, were the development in both dermis and subcutis and the infiltrative growth. Thus, tumor site and pattern of tissue invasion are morphological features that cannot be used to distinguish CPA from HS (Criteria 1 and 2). Endoluminal papillary growth is a well-described morphological aspect of CPA [2] that might assist in the diagnosis of a vascular lesion as reactive rather than neoplastic. Similarly, a multilobular to multifocal growth is more typical of CPA rather than of a true vascular neoplasm (Criterion 3). Importantly, endoluminal papillary growth is not specific of CPA and has been reported in other lesions such as papillary endothelial hyperplasia [3].

Lobular growth is also associated with the presence of interposed normal tissue, a feature that was significantly associated with CPA (Criterion 4). This feature suggests a multifocal disorder rather than a focal expansive growth, thus the identification of adnexal structures between vessels represents a helpful feature for diagnosing CPA versus HS.

In our survey, the simultaneous presence of different vessel types was recorded for most of the CPA cases; therefore, it is a useful parameter, since, in many of the studied cases, capillaries, veins and arteries coexisted often with sparse lymphatics (Criterion 5). Dilated lymphatics in the superficial dermis accompanied by hypoplasia of deep lymphatics and concomitant neovascularization are key features of angiomatosis secondary to lymphedema [2,13]. This entity has been seldom described in dogs [2,13] and resembles congenital angiodysplasia seen in human Klippel–Trenaunay syndrome [14].

Additional interesting observations regarding histological distinctions between CPA and HS included the fact that cytological criteria of malignancy were never observed in CPA, therefore careful search for these features should be warranted (Criterion 6).

Arteriovenous shunts and/or a few malformed blood vessels were detected in over 50% of CPA cases, thus, these features should be considered as relevant adjunctive histopathological features for the diagnosis of CPA (Criterion 7 and 8).

Regarding arteriovenous shunting, this feature was not detected commonly, but we cannot exclude the underestimation of this feature, as shunting is better and should be demonstrated by angiography [15,16].

Arteriovenous angiomatosis has been recently included in the classification of CPA [3]. This entity needs to be differentiated from arteriovenous fistula. Arteriovenous fistula in the skin has generally a iatrogenic origin, including bites, wounds, surgical interventions, and venous catheterization, and is histologically characterized by lesions of small vessels, including intraluminal fibrin thrombi and fibrinoid, necrotizing and leukocytoclastic vasculitis related to the sudden increased pressure reaching the capillary and venous beds [17]. Features of vasculitis were never detected in this case series of canine CPA.

The presence of arteries in CPA in human medicine is a consolidated diagnostic criterion to differentiate arteriovenous malformations from hemangiomas [18]; large size arterial vessels were the predominant type of vessel in a few cases, thus leading to the hypothesis that these were true malformative and not reactive (CPA) vascular lesions.

Necrosis was never detected in CPA and rarely assessed in HS, thus, its presence may be helpful in assisting the pathologist toward the diagnosis of a malignant rather than a reactive process (Criterion 9).

In this case series, 12 dogs had limb involvement, however, only phalanx bone invasion was recorded. Bone invasion is a feature that may occur in CPA as well as in malignant tumors; however, proliferation of well-formed vessels within the bone represent a clue towards diagnosing CPA instead of HS (Criterion 10). Although CPA may occur in every body region, limbs and digits are the most affected, and digital CPA should be considered at higher risk for a progressive invasive behavior [2].

In this study, the finding of nerve bundles is described for the first time in canine CPA. In 13/16 (81%) cases where arteriovenous shunt or malformed vessels were seen, nerve bundles were also detected among vessels. The finding of thin nerve bundles only in CPA adds a microscopic feature useful for differentiating CPA from well-differentiated HS or other tumors (Criterion 11).

As emerging from previous caseloads, this study confirms that data on signalment, clinical presentation, and follow up may support histopathological diagnosis, but are not sufficient to differentiate CPA from other lesions, including neoplasia. Because of the paucity of fully documented cases, it is not possible to establish breed or gender predisposition for canine CPA. Almost half of the cases in this work were from mixed breed dogs (41.9%), paralleling previous reports where mainly mixed breeds are affected by CPA. Reported breeds include Labrador retriever, Shar-pei and, Cocker spaniel [1,2,4,5,6,7,8,9].

Noteworthy, in this study, six dogs were 3 years of age or younger; the young age at presentation might represent a useful clue for alerting the pathologist that the lesion represents a non-neoplastic vascular disorder. However, as older dogs were affected by CPA, diagnostic imaging and skin biopsies should be always included in the diagnostic workup when a cutaneous vascular disorder is suspected.

In this caseload, almost half of the dogs presented lesions on the limbs, paralleling reported sites of emergence of CPA in previous publications [2,4,5,7,9,10]. Most common sites of occurrence include distal extremities (digits and feet) and the head, although other anatomical sites may be involved [1,2,6]. On the contrary, feet were not a common site of CPA emergence in this caseload, and this may account for the lack of bony invasion reported in this caseload.

Although out of the scope, the main limit of this study regards the follow up data collection obtained only from five cases. However, no history of metastatic disease or death related to disease were recorded.

Noteworthy, malformed vessels were detected in a subgroup of canine CPA cases. This finding in association with the young age (sometimes appearing at birth) of many cases suggest that some forms of CPA may evolve from a congenital lesion (malformative origin) that cannot be readily detected at birth or soon thereafter because of the deep tissue location. Noteworthy, it is well known in human medicine that those cases of vascular malformations located in deep tissues become evident later in life upon progression of disease, because of the abnormal blood pressure/flow inducing progression of the disease. This hypothesis needs to be further investigated in a larger caseload in association with adequate diagnostic imaging procedures.

In conclusion, although CPA has been clinically and histologically described previously in dogs, this study represents the first attempt to detail CPA histological features and their significance in the diagnosis of this entity. Several features have demonstrated to be highly useful for CPA identification against HS. Further studies and complete clinical descriptions are needed to confirm our findings and to verify the malformative or proliferative origin of CPA, and should include clinical presentation, diagnostic imaging confirmation, histomorphological aspects, natural evolution, and therapy response.

## Figures and Tables

**Figure 1 vetsci-09-00340-f001:**
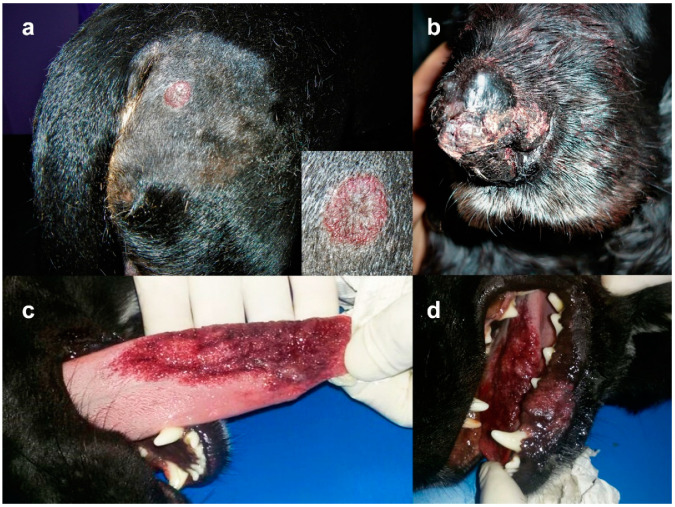
Clinical presentation of CPA. (**a**) Case N 9: round alopecic and erythematous plaque lateral to the tail base (detail of the lesion in the inset); (**b**) Case N 16: multiple hemorrhagic, crusted, confluent plaques and nodules on the muzzle; (**c**) Case N 17: large hemorrhagic plaques on the inferior lip and tongue (Courtesy dr. Paolo Persico); (**d**) Case N 17: a closer view of the hemorrhagic plaque on the tongue.

**Table 1 vetsci-09-00340-t001:** Data on signalment, type and distribution of lesions (m = month, y = year, na = not available).

Case N°	Breed	Age	Gender	Site	Type of Lesion
1	Mixed	7 y	F	Limb, elbow	na
2	Springer spaniel	10 y	M	Limb, carpus (volar)	Nodule, violet, not painful, 2–3 m lasting
3	Mixed	3 y	M	Limb, anterior, thigh	Ulcerated, 1 y lasting
4	Mixed	3 y	M	Limb, metatarsus	Nodule, pink-red, 1 y lasting
5	Mixed	11 y	M	Limb, lateral thigh	Nodule, red
6	Bull mastiff	8 y	F	Limb, carpus, flank	Nodule 3 cm on carpus multiple on flank, 2 m lasting
7	Dogue de Bordeaux	6 m	M	Limb, lateral knee	Nodule, 2 cm, at birth
8	Yorkshire terrier	2 y	M	na	na
9	Dachshund	9 y	M	Tail base, lateral	Plaque, pink-red, alopecic, 1 cm, 4 m lasting
10	Dogue de Bordeaux	1 y	M	Head, cheekbone	3 cm, serpiginous, 1 m lasting
11	Mixed	3 y	F	Hind limb	na
12	Mixed	na	na	Head, ear, bilateral	na
13	Boxer	6 y	M	Sternum	Nodule, red, ulcerated, since long time
14	Dachshund	2 m	F	Head, lip	Plaque, erythematous, 2 cm, at birth
15	Mixed	10 y	M	Limb, metatarsus	Nodule, peduncolated, 1 m lasting
16	Cocker spaniel	10 y	F	Head, muzzle	Multiple nodules, confluent, 5 y lasting
17	Mixed	1 y	M	Top of the head, lip margin, tongue	Multiple plaques, erythematous, crusted, 2 ms lasting
18	German shepherd	6 y	M	Limb, tarsum	Nodule, simil callus, 3 cm
19	Mixed	11 y	M	Abdomen, peripreputial	Multiple red plaques, 0.5 cm
20	Mixed	na	M	na	na
21	Mixed	5 y	F	Ear margin	na
22	Shorthair pointer	na	na	na	na
23	Husky	13 y	F	Knee	Associated with SCC
24	Mixed	8 y	M	Sternum	Papillated, ulcerated
25	Pincher	4 y	M	Ear, bilateral, hemorrhagic	na
26	Mixed	9 y	M	Sternum	Nodule, ulcerated, hemorrhagic
27	Australian sheepdog	4 y	M	Dorsum	Nodule, ulcerated
28	Great Dane	6 y	M	Forelimb (3rd digit)	Subcutaneous mass

## Data Availability

Not applicable.

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
