# Peer review of "Identification of Histopathological Criteria for the Diagnosis of Canine Cutaneous Progressive Angiomatosis"

_vetsci, 2022, doi:10.3390/vetsci9070340_

Round 1

Reviewer 1 Report

L52 prognosis depending on the extension and  depth of the infiltration of tissue invasion. 

AND LOCALIZATION … for ex. lesions on the paw is really difficult to manage 

L79 and reviewed by a pathologist with dermatopathol- 

If I well understand …reviewed by 4 pathologists independantly… (or one case was review by one pathologist another one by an other ?)

Table 1 case N°7 it is DOGUE de B not Dougue de Bordeaux 

Excellent paper but to me,  the title could be changed to emphasize the criteria importance (that’s really helpful and new)

Something like this?

Identification of histopathological criteria to assess the diagnostic of Canine cutaneous progressive angiomatosis

Author Response

Comments and Suggestions for Authors from reviewer 1

  • L52 prognosis depending on the extension and  depth of the infiltration of tissue invasion. AND LOCALIZATION … for ex. lesions on the paw is really difficult to manage 

OK amended (Line 64-65)

  • L79 and reviewed by a pathologist with dermatopathol-  If I well understand …reviewed by 4 pathologists independantly… (or one case was review by one pathologist another one by an other ?)

OK we are sorry for confusion, two pathologists with dermatopathology expertise reviewed the cases (AB and PR), the text has been corrected (Line 92).

  • Table 1 case N°7 it isDOGUE de B not Dougue de Bordeaux 
    OK amended

  • Excellent paper but to me,  the title could be changed to emphasize the criteria importance (that’s really helpful and new). Something like this? Identification of histopathological criteria to assess the diagnostic of Canine cutaneous progressive angiomatosis.
    OK we changed the title with the one you proposed.

During the revision procedure we realized that the last three references listed were object of study for the authors but were not cited in the text. We erased them from the reference list.

Reviewer 2 Report

The paper submitted by Abramo et al. provides essential and valuable information about this condition in dogs. Moreover, the authors give helpful criteria to differentiate it from other diseases.

Before acceptance, I have provided some specific comments

In the abstract, Please explain better the last sentence. Some gross pictures should be included. In figure 2, (f) please add a single arrow in the mitotic figure. In lines 223-225, please clarify the statement. In the discussion section, I will try to join all the key findings. Please, could you be more precise about the diagnostic imaging procedure to assess CPA (I suppose CT or ultrasound with Doppler). Line 269, please explain “destructive behaviour”. Please, avoid “Noteworthy” repetition.

Author Response

Comments and Suggestions for Authors from reviewer 2

The paper submitted by Abramo et al. provides essential and valuable information about this condition in dogs. Moreover, the authors give helpful criteria to differentiate it from other diseases.

Before acceptance, I have provided some specific comments

  • In the abstract, Please explain better the last sentence.

OK the phrase “In conclusion, presence and/or absence ……. for CPA diagnosis” has been changed with “In conclusion, the combined use of different microscopical clues allowed for the distinction of CPA from HS and was considered useful for the diagnosis of CPA” (Lines 45-48).

  • Some gross pictures should be included.

OK Figure 1 with clinical presentation of lesions in three dogs has been added (Line 139-143).

  • In figure 2, (f) please add a single arrow in the mitotic figure.
  1. Amended.

  • In lines 223-225, please clarify the statement.

We do not well understand this question. The statement is extensively clarified in the following lines, from 247 to 255.

  • In the discussion section, I will try to join all the key findings.
  1. We specified for each paragraph the number of criterion discussed to assist the reader. We also add comment to Criterion 9 that was lacking (Lines 282-284).

  • Please, could you be more precise about the diagnostic imaging procedure to assess CPA (I suppose CT or ultrasound with Doppler).

OK angiography is the preferred procedure to evaluate normal or aberrant vascular plexus, it has already used and described in the literature, however most of the cases refer to cats. We will add name of the procedure and some of the references that can be used to become familiar with the procedure (Line 269).

  • Line 269, please explain “destructive behaviour”.

The term “destructive” has been changed with “invasive” (Line 307)

  • Please, avoid “Noteworthy” repetition.
  1. Erased (Line 276).

During the revision procedure we realized that the last three references listed were object of study for the authors but were not cited in the text. We erased them from the reference list.
